# ATR Inhibition Potentiates PARP Inhibitor Cytotoxicity in High Risk Neuroblastoma Cell Lines by Multiple Mechanisms

**DOI:** 10.3390/cancers12051095

**Published:** 2020-04-28

**Authors:** Harriet E. D. Southgate, Lindi Chen, Deborah A. Tweddle, Nicola J. Curtin

**Affiliations:** 1Wolfson Childhood Cancer Research Centre, Newcastle Centre for Cancer, Translational and Clinical Research Institute, Faculty of Medical Sciences, Newcastle University, Newcastle Upon Tyne NE1 7RU, UK; 2Newcastle Centre for Cancer, Translational and Clinical Research Institute, Faculty of Medical Sciences, Newcastle University, Newcastle Upon Tyne NE2 4HH, UK

**Keywords:** poly (ADP-ribose) polymerase inhibitors (PARPi), ataxia telangiectasia and Rad3 related inhibitors (ATRi), replication stress, cell cycle checkpoints

## Abstract

*Background:* High risk neuroblastoma (HR-NB) is one the most difficult childhood cancers to cure. These tumours frequently present with DNA damage response (DDR) defects including loss or mutation of key DDR genes, oncogene-induced replication stress (RS) and cell cycle checkpoint dysfunction. Aim: To identify biomarkers of sensitivity to inhibition of Ataxia telangiectasia and Rad3 related (ATR), a DNA damage sensor, and poly (ADP-ribose) polymerase (PARP), which is required for single strand break repair. We also hypothesise that combining ATR and PARP inhibition is synergistic. *Methods:* Single agent sensitivity to VE-821 (ATR inhibitor) and olaparib (PARP inhibitor), and the combination, was determined using cell proliferation and clonogenic assays, in HR-NB cell lines. Basal expression of DDR proteins, including ataxia telangiectasia mutated (ATM) and ATR, was assessed using Western blotting. CHK1^S345^ and H2AX^S129^ phosphorylation was assessed using Western blotting to determine ATR activity and RS, respectively. RS and homologous recombination repair (HRR) activity was also measured by γH2AX and Rad51 foci formation using immunofluorescence. *Results:*
*MYCN* amplification and/or low ATM protein expression were associated with sensitivity to VE-821 (*p* < 0.05). VE-821 was synergistic with olaparib (CI value 0.04–0.89) independent of *MYCN* or ATM status. Olaparib increased H2AX^S129^ phosphorylation which was further increased by VE-821. Olaparib-induced Rad51 foci formation was reduced by VE-821 suggesting inhibition of HRR. *Conclusion:* RS associated with *MYCN* amplification, ATR loss or PARP inhibition increases sensitivity to the ATR inhibitor VE-821. These findings suggest a potential therapeutic strategy for the treatment of HR-NB.

## 1. Introduction

Inhibitors of ataxia telangiectasia and Rad3 related (ATR) kinase are currently being tested in early phase clinical trials for patients with adult cancers. ATR is a DNA damage sensor kinase which has a pivotal role in recovery from replication stress (RS). Activated ATR signals to many pathways involved in regulation of origin firing, and stabilisation and restart of stalled replication forks (reviewed in: [1,2]). ATR signals to S and G2/M cell cycle checkpoint control by inhibiting CDK2 and CDK1 via its primary target CHK1 [3,4]. ATR inhibition abrogates S and G2/M checkpoint arrest, leading to cell death by mitotic catastrophe [5,6].

Two ATR inhibitors, AZD6738 (Astra Zeneca) and M6620 (formerly VX-970, Merck), have entered phase II trials, with a third, BAY1895344 (Bayer) in phase I (listed on https://clinicaltrials.gov/). Various determinants of ATR inhibitor sensitivity have been proposed, including RS, loss of G1 checkpoint control and loss of the ataxia telangiectasia mutated (ATM) protein [7,8].

RS is a common feature of cancer cells and leads to genetic instability. There are many causes of RS including overexpression of proliferation-driving oncogenes (*Ras, Myc, cyclin E* etc.), deregulation of replication origin firing, limited nucleotide pools or essential replication factors and replication through fragile sites or damaged DNA regions [9,10].

Loss of G1 checkpoint control also contributes to RS and is common in cancer through loss of tumour suppressors such as p53, pRB and ATM, imbalance of cyclins, cyclin-dependent kinases and their inhibitors and expression of oncogenes [11]. G1 checkpoint deficiency results in a reliance on the S and G2/M checkpoints to maintain genome integrity and prevent replication of damaged DNA/mitotic catastrophe [12,13,14].

Neuroblastoma (NB) is a rare embryonal tumour derived from cells of the developing sympathetic nervous system. Around 100 cases are diagnosed a year in the UK, of which 50% are classified as high risk, but accounts for ~10% of paediatric cancer deaths [15,16]. Long term survival of high-risk neuroblastoma (HR-NB) (metastatic disease over 1 year of age- or *MYCN*-amplified disease) currently remains less than 50% at 5 years despite intensive high-dose multimodal treatment [17,18]. Survival of relapsed NB is particularly poor with less than 10% 5-year survival [19]. HR-NB frequently present with DNA damage response (DDR) defects including loss or mutation of key DDR genes, oncogene-induced RS and cell cycle checkpoint dysfunction, which suggest they would be sensitive to ATR inhibition [20,21,22,23].

Fifty percent of HR-NB have amplification of the *MYCN* oncogene, leading to RS. *MYCN*-amplified (MNA) tumours also show defective G1 checkpoint arrest [23]. A common genetic abnormality observed in non-MNA NB is allelic loss of chromosome 11q. Tumours with 11q deletion display a poor prognosis similar to MNA [24]. Many genes coding for proteins involved in the DDR are located on 11q including *ATM*, *CHEK1*, *H2AFX* and *MRE11* [20]. Together, MNA and 11q deletion occur in 70–80% of HR-NB tumours.

Although rare at diagnosis, defects in p53 signalling have been observed in up to 50% relapsed NB tumours [22,25], causing further G1 checkpoint dysfunction and abrogating the p53 dependent intrinsic apoptosis pathway.

Poly ADP-ribose polymerase (PARP) inhibitors also cause RS [26]. PARP is activated in response to DNA single strand breaks and orchestrates repair [27]. Several PARP inhibitors have been approved for ovarian and breast cancer with defective homologous recombination repair. There are currently seven clinical trials testing the use of PARP inhibitors for paediatric tumours of which only three include NB (https://clinicaltrials.gov/: NCT04236414, NCT03233204, NCT02392793).

Preclinical testing of the PARP inhibitor olaparib (Astra Zeneca) in NB shows that PARP inhibition potentiates the cytotoxic effect of a variety of chemotherapy agents and ionising radiation [28,29,30,31]. In addition, NB tumours with *MYCN* amplification or *ATM* deficiency have been shown to have increased sensitivity to single agent olaparib treatment [32,33].

We aimed to test if the DDR defects frequently observed in NB would be potential predictive biomarkers of sensitivity to ATR inhibition using VE-821 (the preclinical lead from which M6620 was developed).

We hypothesise that there will be mutual synergy between ATR and PARP inhibitors by further increasing RS, irrespective of *MYCN* or *ATM* status, by the accumulation of unrepaired single strand breaks, when PARP is inhibited, and failure to arrest in S-phase when ATR is inhibited.

In this study, we identify features of NB cell lines that determine sensitivity to ATR inhibition, for use as potential predictive biomarkers, and examine the effect of ATR inhibition on the cytotoxicity of the PARP inhibitor olaparib.

## 2. Results

### 2.1. DDR Protein Expression in NB Cell Lines

To reflect the variety of DDR defects observed in NB tumours, we chose a panel of NB cell line of varying *MYCN*, 11q and *TP53* status to interrogate what features would lead to sensitivity to ATR and PARP inhibitors. The genetic features of these cell lines are listed in Table 1.

We analysed endogenous expression of key DDR proteins (ATM, ATR, CHK1, CHK2, MYCN and p53) in these cell lines as well as baseline activity of ATR, ATM (phospho-CHK1^S345^ and phospho-CHK2^T68^ and expression, respectively) by Western blot (Figure 1A). Mean protein band intensities of MYCN, ATM and p53 from two independent experiments are shown in Figure 1B. ATM and p53 function after activation with doxorubicin was also examined (Figure 1C,D, respectively).

As expected, *MNA* cell lines show high MYCN protein expression compared to non-*MNA* cell lines (Figure 1A and Appendix A), with the exception of SJNB1, which has high expression of MYCN in the absence of a gene amplification. In contrast, some cell lines with 11q deletion have baseline ATM expression, suggesting that ATM expression from the other allele is sufficient to produce a functional protein (Figure 1A,C). Cell lines with *TP53* mutations show stabilised p53 protein (NMB and Kat100) or no p53 protein expression (SKNAS and IGRN91). The *TP53* mutation in the NMB and Kat100 cell lines are point mutations leading to accumulation and stabilisation of the dysfunctional protein (Figure 1D and previously in [38,41]), whereas SKNAS and IGRN91 have a deletion and duplication, respectively, of whole exons and do not stabilise the protein. The IGRN91 cell line expresses a high molecular weight gene product after activation with doxorubicin (consistent with duplication of exons 7–9 [35]), which retains some function (Figure 1D).

### 2.2. MYCN Amplification and Low ATM Expression are Determinants of Sensitivity to ATR Inhibition in NB Cell Lines

Growth inhibition by the ATR inhibitor VE-821 was determined by XTT cell proliferation assay (Figure 2A). Cell lines were grouped based on genetic features, baseline MYCN, p53 and ATM protein expression above (high) or below (low) median expression (Figure 1B) and ATM and p53 response to doxorubicin. Growth inhibition at 10 μM for each cell line was analysed across groups (Figure 2B–D) by Mann–Whitney *U* test. *MNA* cell lines show high MYCN protein expression (Figure 1B) and these cell lines were significantly more sensitive to 10 μM VE-821 than non-*MNA* cell lines (*p* < 0.05). Although there was no significant difference in VE-821 sensitivity between cell lines with or without *ATM* aberration (*ATM* mutation and/or 11q deletion), cell lines with low baseline ATM protein expression were significantly more sensitive than cell lines with high baseline ATM expression (*p* < 0.05). In regression analysis, MYCN and ATM protein expression was negatively and positively correlated with sensitivity to VE-821, respectively (Appendix A). Cell lines with dysfunctional ATM were more sensitive to VE-821 than cell lines with functional ATM, indicated by ATM^S1981^ auto-phosphorylation after treatment with 1 μM doxorubicin (Figure 1C). There was no significant difference in sensitivity to VE-821 when cell lines were grouped according to *TP53* mutation status, p53 protein expression or p53 function. These results were confirmed by clonogenic survival assay (Appendix A).

### 2.3. MYCN Amplification and Low ATM Expression Are Not Significant Determinants of Sensitivity to PARP Inhibition in NB Cell Lines

To see if the features that are associated with ATR inhibitor sensitivity also determine sensitivity to PARP inhibition, sensitivity to the PARP inhibitor olaparib was determined by XTT cell proliferation assay (Appendix A). Cell lines were grouped by the molecular features above and growth inhibition at 10 μM olaparib for each cell line was analysed. Unlike for VE-821, there was no significant difference in olaparib sensitivity when cell lines were grouped according to *MYCN* amplification, MYCN protein expression, ATM protein expression or ATM function (Appendix A). There was no significant difference in sensitivity to olaparib when cell lines were grouped according to *TP53* mutation status, p53 protein expression or p53 function. Single agent sensitivity of cell lines to VE-821 and olaparib by XTT cell proliferation assay is summarised in Table 2.

However, when analysed by clonogenic assay (Appendix A), cell lines with high MYCN protein expression were significantly more sensitive to olaparib (*p* < 0.05). This was not the case when cell lines were analysed based on *MYCN* amplification and some cell lines, such as SJNB1, which are not *MNA* but express high levels of MYCN, were relatively sensitive to olaparib.

### 2.4. PARP and ATR Inhibition Synergistically Inhibit Cell Growth

To test if growth inhibition by olaparib and VE-821 is synergistic, an XTT cell proliferation assay was carried out testing the response of four NB cell lines to increasing concentrations of olaparib with the addition of fixed concentrations of VE-821. SHSY5Y (non-MNA, p53 wt), SKNAS (non-MNA, p53 mutant), NGP (MNA, p53 wt) and N20_R1 (MNA, p53 mut) cells were treated with 0, 0.1, 1, and 10 µM olaparib with the addition of either 0, 0.1 or 1 μM VE-821 (Figure 3A). VE-821 significantly sensitized the SKNAS, NGP and N20_R1 cell lines to olaparib, fold sensitisation 1.43-4.60 (Figure 3B,C). Although not statistically significant, the cytotoxicity of olaparib was increased by VE-821 in the SHSY5Y (non-*MNA*, *TP53* wt) cell line, fold sensitisation 1.41. Synergism between VE-821 and olaparib was determined by combination index analysis (calcusyn) where combination index (CI) <1 indicates synergy. PARP inhibition was synergistic with ATR inhibition in all cell lines, irrespective of *MYCN* or *TP53* status (Figure 3D). This was also confirmed by clonogenic survival assay (Appendix A).

### 2.5. ATR Inhibition Increases Replication Stress Caused by Olaparib by Blocking S and G2 Cell Cycle Arrest and Reducing Homologous Recombination Repair

PARP inhibition has been shown to increase RS in NB cell lines [32]. Since ATR inhibition potentiated olaparib induced growth inhibition, we investigated the effect of ATR and PARP inhibitor combination on RS markers. As most markers of RS are dependent on ATR activity, such as CHK1^S345^ and RPA2^T21^ phosphorylation, we measured phosphorylation of histone 2AX (γH2AX) as a surrogate marker of RS. γH2AX is a marker of DNA double strand breaks (DSB) and RS. In the absence of genotoxic agents, γH2AX primarily marks RS. SHSY5Y, SKNAS, NGP and N20_R1 cells were treated with olaparib 5 μM and/or VE-821 1 μM for 24 h. pCHK1^S345^ (ATR activation), pCHK2^T68^ (ATM activation) and γH2AX (RS) protein expression was measured by Western blot (Figure 4A). Protein expression was quantified by densitometry (ImageJ) and compared to DMSO control (Figure 4B). Treatment with 5 μM olaparib resulted in activation of ATR (pCHK1^S345^) in all cell lines, which was reduced with the addition of 1 μM VE-821 as expected. pCHK2^T68^ and γH2AX expression increased after treatment with olaparib and was further increased with the addition of VE-821, suggesting increased RS and DNA damage. Another marker of RS is phosphorylation of RPA2^S4/S8^ by ATM and DNA-PK [45]. Using the SHSY5Y cell line, expression of pRPA2^S8^ was measured after treatment with olaparib 5 μM +/- VE-821 1 μM over the course of 48 h (Figure 4C). The addition of VE-821 1 μM brought forward the RPA2^S8^ phosphorylation induced by olaparib 5 μM by 24 h.

ATR signals to S and G2 cell cycle arrest [4], which are abrogated when ATR is inhibited [5,6,46]. Since olaparib is known to lead to cell cycle arrest [47,48], we investigated the effect of olaparib and VE-821 combination on the cell cycle in SHSY5Y, SKNAS and NGP cell lines (Figure 5A). Olaparib treatment alone increased the proportion of cells in S and G2 phase for each cell line, consistent with response to olaparib-induced RS. This cell cycle arrest was not seen with the addition of both inhibitors, suggesting loss of S and G2 checkpoints arrest.

We also examined the effect on homologous recombination repair (HRR). PARP inhibition is synthetically lethal with HRR deficiency [49]. ATR inhibition has been shown to reduce Rad51 foci formation (an indicator of HRR activity [50]), inducing a HRR-deficient phenotype [51,52]. We analysed the effect of 10 μM olaparib and/or 1 μM VE-821 on Rad51 and γH2AX foci in SHSY5Y, SKNAS, NGP and N20_R1 cell lines. Representative γH2AX and Rad51 foci images from SKNAS (non-*MNA*, *TP53* mutant) and NGP (*MNA*, *TP53* wt) cell lines treated with 1 µM VE-821, 10 µM olaparib or both are shown in Figure 5B. Treatment with olaparib induced an increase in Rad51 foci formation, which was suppressed with the addition of VE-821 in all cell lines except SKNAS (Figure 5C). Interestingly, SKNAS cells are relatively resistant to both olaparib and VE-821. RS was measured by γH2AX nuclear fluorescence intensity, which increased after treatment with olaparib and increased further after treatment with both inhibitors in all cell lines (Figure 5D).

Overall, the combination of olaparib and VE-821 leads to increased RS due to defective S and G2 cell cycle arrest and suppression of HRR leading to collapsed replication forks.

## 3. Discussion

It is widely accepted that overexpression of proliferation-inducing oncogenes leads to RS and dependency on ATR signalling [53]. In models of Ras- or MYC-driven cancer, signalling through these oncogenes has been shown to lead to sensitivity to ATR inhibition [54,55,56]. In NB cell lines, high MYCN expression has been shown to increase RS [32,57]. Here we have shown that amplification and overexpression of the *MYCN* oncogene in NB cell lines is associated with sensitivity to VE-821, confirming that *MYCN* driven NB cells are vulnerable to ATR inhibition.

ATM loss or dysfunction is also associated with sensitivity to VE-821. Our results confirm that reduced ATM function, which can be partially determined by protein level, confers sensitivity to ATR inhibition in NB as it does in other tumour types [5,58,59,60]. In addition to 11q deletion, loss of ATM can occur in *MNA* NB by upregulation of the ATM targeting micro-RNA, miR-421 [61]. ATM silencing, in addition to MYCN-driven proliferation, is likely contributing to replication stress through impaired DNA DSB signalling and repair, making these cells especially vulnerable to ATR inhibition.

MYCN and ATM protein expression levels were more powerful determinants of VE-821 sensitivity than the genetic status (amplification or deletion/mutation, respectively), and MYCN protein levels positively correlated with VE-821 sensitivity, whereas ATM protein level was negatively correlated with VE-821 sensitivity.

Mutation, expression or dysfunction of the p53 tumour suppressor protein was not associated with sensitivity to ATR inhibition. Middleton et al. also demonstrated that defective p53 signalling did not lead to sensitivity to VE-821 alone, but the addition of VE-821 showed greater potentiation of gemcitabine and ionising radiation induced cytotoxicity in p53^-/-^ cell lines [46].

To summarise, MYCN and ATM protein expression levels, as well as genetic status, could provide useful predictive biomarkers to stratify NB patients who would benefit from an ATR inhibitor.

Further investigation into combining ATR inhibition with DNA-damaging chemotherapy may provide a novel therapeutic strategy for both *TP53* wt and *TP53* mutant NB.

In contrast, neither *ATM* nor *MYCN* status significantly determined sensitivity to PARP inhibition by olaparib. Colicchia et al. previously demonstrated that *MYCN*-amplification or overexpression renders NB cells especially sensitive to single agent olaparib [32]. Sensitivity of cell lines to PARP inhibitors by this group was analysed by MTS cell proliferation assay, an assay which measures metabolic activity in a similar way to XTT. Although we did not see a significant difference between MNA cell lines and non-MNA cell lines in olaparib sensitivity in either XTT or clonogenic survival assay, high MYCN expressing cell lines did show significantly increased sensitivity when analysed by clonogenic survival. The clonogenic survival assay measures the ability of single cells to divide indefinitely and relies on cell lines having good cloning efficiency. Unfortunately, some NB cell lines show poor cloning efficiency and some cell lines show stark differences in sensitivity between the two assays.

Our results highlight the importance of validating sensitivity found to novel agents in models of disease with other types of cell viability assays. In the case of NB, more evidence is required before the role of MYCN in PARP sensitivity can be established. It may be that in the context of complex genetic alterations, cells may have acquired a spectrum of features leading to resistance, as well as sensitivity to these inhibitors, making it difficult to predict response.

In addition to investigating features associated with sensitivity to these inhibitors as single agents, we provide evidence of synergy between ATR and PARP inhibition in NB cell lines. VE-821 enhanced olaparib-induced growth inhibition independently of *MYCN*, *ATM* and *TP53* status. ATR inhibition has been shown to enhance PARP inhibitor sensitivity in a variety of adult solid tumour types [51,62,63] and overcome resistance to PARP inhibitors [52,64].

Olaparib treatment alone induced markers of RS (γH2AX, pRPA2^S8^, Figure 4 and Figure 5) and activated ATR (pCHK1^S345^ expression, Figure 4A). When ATR was inhibited, olaparib-induced markers of RS were increased, which suggested the inhibition of ATR exacerbated RS caused by PARP inhibition. This could be because inhibition of ATR releases olaparib-induced S and G2 arrest, shown here (Figure 5A) and previously reported [63]. Loss of cell cycle checkpoints results in DNA replication through damaged areas, fork stalling, collapse and chromosome breakage, which, if left unrepaired, leads to mitotic catastrophe and cell death [44].

Loss or inhibition of ATR has been shown to restrict HRR function [51,52,65]. We confirmed this in MNA and non-MNA NB cell lines. However, one non-MNA cell line, SKNAS, did not show increased Rad51 after PARP inhibition, and would be considered HRR-defective. HRR-defective cells are usually highly sensitive to PARP inhibitors, whereas the SKNAS cell line is the most resistant to olaparib out of the cell lines analysed in this study.

Although inhibiting ATR to specifically target RS in cancer cell looks promising, other mechanisms of tolerating RS have been identified, such as overexpression of components of the fork protection complex [66] and recruitment of Y-family (translesion synthesis) DNA polymerases [67,68], which should be considered as possible resistance mechanisms. These are all factors to be considered when interpreting data from the recently initiated trials of PARPi and ATRi combinations (NCT02723864, NCT03462342, NCT04065269, NCT03787680, NCT04149145, and NCT03682289).

In conclusion, we have shown that the combination of ATR and PARP inhibition leads to increased RS and cytotoxicity in NB cell lines. This synergy is likely to be, at least in part, not only due to abrogation of cell cycle checkpoints but also inhibition of HRR, thereby inducing synthetic lethality with PARP inhibition.

## 4. Materials and Methods

### 4.1. Chemicals and Cell Lines

VE-821 and Olaparib were purchased from Stratech Scientific Ltd. (Cambridge, UK) and stored at −20 °C in stock solution of 100 mM and 20 mM, respectively, in DMSO (Sigma-Aldrich, Gillingham, Dorset, UK).

Neuroblastoma cell lines used in this study were: SHSY5Y, SKNAS, NGP, N20_R1, NMB, IMR32, Kat100, IGRN91, SJNB1, and GIMEN (Table 1). All neuroblastoma cell lines were obtained between 1996 and 2018, with the exception of N20_R1, and were validated upon receipt using cytogenetic analysis courtesy of Dr Nick Bown (Institute of Genetic Medicine, Newcastle University). N20_R1 (N_N20R1) were generated from parental NGP cell line with resistance to Nutlin-3 [37]. All cell lines were cultured in RPMI-1640 (Sigma-Aldrich) supplemented with 10% (*v*/*v*) Fetal Calf Serum (Gibco, Life Technologies Ltd., Thermo Fisher Scientific, Waltham, MA, USA) and maintained at 37 °C in a humidified incubator with 5% CO_2_. Cell lines were routinely tested for Mycoplasma and were confirmed to be negative.

### 4.2. Growth Inhibition Assay

Cells were seeded in 96-well plates (Corning, VWR International Ltd., Lutterworth, UK), and allowed to adhere overnight before treatment with ATR or PARP inhibitors alone or in combination for 72 h. Inhibitors were added at 200× dilution to give a final DMSO concentration of 0.05%. Percentage control growth was assessed using the XTT cell proliferation assay (Roche, Burgess Hill, UK) according to the manufacturer’s instructions and using the following formula: (average absorbance test/average absorbance control) × 100. Combination Index (CI) values were determined by the Chou–Talalay method using CalcuSyn v2 (Biosoft, Cambridge, UK).

### 4.3. Protein Extraction and Western Blotting

Cell pellets were harvested and protein extracted using PhosphoSafe™ Extraction Buffer (Novagen, Merck Millipore Ltd. Watford, UK) following the manufacturer’s protocol. Protein concentration was quantified using the Pierce BCA protein assay kit (Thermo Fisher Scientific).

Proteins were separated using 3–8% Criterion™ XT Tris-Acetate Protein Gel (Bio-Rad Laboratories Ltd., Hemel Hempstead, UK) and transferred onto Hybond-C Extra membrane (GE Life Sciences, Little Chalfont, UK). Membranes were stained with Ponceau S (Sigma-Aldrich) to control for loading, destained in tris buffered saline, 0.5% tween 20 (TBST) and blocked for 1 h in 5% milk TBST.

Primary antibodies were: mouse monoclonal MYCN 1:1000 (Santa Cruz Bio-technology (SCBT), Dallas, TX, USA: A1513), rabbit polyclonal ATM 1:500 (cell signalling technology(CST), Danvers, MA, USA: 2873S), mouse phospho-ATM (ser1981) 1:1000 (CST: 4526), mouse monoclonal CHK1 1:1000 (SCBT: sc-8408), mouse monoclonal CHK2 1:1000 (SCBT: sc-17747), rabbit polyclonal phosho-CHK1 (ser345) 1:1000 (CST:2341S), rabbit monoclonal phosphor-CHK2 (thr68) 1:1000 (CST:2197S), rabbit polyclonal ATR 1:500 (CST: 2790S), p53 1:1000 (NCL-L-p53-DO7, Leica Biosystems Ltd.), rabbit monoclonal p21 Waf1/Cip1 (12D1) 1:1000 (CST: 2947S), rabbit monoclonal phosopho-H2AX (ser139) 1:2000 (CST: 2577S), rabbit monoclonal phospho-RPA2 (ser8) 1:1000 (CST: 54762S), mouse monoclonal glyceraldehyde 3-phosphate dehydrogenase (GAPDH) 1:5000 (SCBT: sc-47724). Secondary antibodies used were peroxidase-conjugated goat anti-mouse (Dako, Glostrup, Denmark) and anti-rabbit immunoglobulins (Dako) at 1:2500.

Protein detection was performed using enhanced chemiluminescence (Bio-Rad) and imaged using the ChemiDoc imaging system (Bio-Rad Laboratories Ltd., Hemel Hempstead, UK). Densitometry was performed using ImageJ image analysis software (Version 1.52p; Java 1.8.0_172 [64-bit]).

### 4.4. Cell Cycle Analysis

Cells were harvested post-treatment, fixed in ice-cold 70% (*v*/*v*) ethanol and stored at −20 °C. Prior to analysis, cells were washed with phosphate buffered saline (PBS), resuspended in 500 μL PBS with 50 μg/mL propidium iodide (Sigma-Aldrich) and 50 μg/mL RNAse A (Sigma-Aldrich), and incubated at 37 °C for 30 min. Samples were analysed on the Attune NxT Flow Cytometer using Invitrogen™ Attune NxT Software (Thermo Fisher Scientific). Data were analysed using FlowJo ^TM^ (BD Biosciences, Wokingham, UK). Experiments were at least *n* = 3.

### 4.5. Immunofluorescence

Cells were treated for 24 h with control vehicle (DMSO) and 1 μM VE-821 with or without 10 μM olaparib. Cells were stained with mouse monoclonal anti phospho-Histone H2A.X (Ser139) antibody (SCBT) at 1:500 and rabbit monoclonal anti RAD51 antibody (CST: 8875S) at 1:250. Secondary antibodies used were Alexa 488 conjugated goat anti rabbit and Alexa 546 conjugated goat anti mouse (Invitrogen, Thermo Fisher Scientific), both at 1:1000. Cells were imaged using a Leica DM6 microscope and Leica Application Suite (LAS) X software (Leica Microsystems, Wetzlar, Germany). The number of RAD51 foci in each cell and total nuclear fluorescence intensity for γH2AX were quantified using ImageJ software (Version 1.52p; Java 1.8.0_172 (64-bit)) and data was plotted using GraphPad Prism v6 (San Diego, CA, USA).

### 4.6. Statistical Analysis

Mann Whitney U and 2-way ANOVA statistical tests were carried out using GraphPad Prism v6 software and *p* < 0.05 was taken to be statistically significant.

## 5. Conclusions

In the era of precision medicine, cancer specific DDR defects and RS have become attractive targets, with many novel agents entering clinical trials. With the introduction of PARP inhibitors into clinical trials for the treatment of paediatric cancers, including HR-NB, a disease with one of the worst long term prognoses, and ATR inhibitors being tested in adult trials, we sought to investigate which NB specific DDR defects, if any, would lead to sensitivity to these agents alone and in combination. We have shown that *MYCN*-amplification and protein overexpression and loss of functional ATM protein through deletion or mutation result in vulnerability to ATR inhibition by VE-821. The case for PARP inhibition is less clear and, although there is a trend towards MYCN overexpressing cells being more sensitive to olaparib, the role of MYCN in PARP inhibitor sensitivity needs to be investigated further. We also provide evidence of synergy between ATR and PARP inhibition in NB cell lines independent of *MYCN*, *ATM* or *TP53* status. This may be due to increased RS, compromised S and G2 checkpoint arrest and/or defective HRR in the presence of both PARP and ATR inhibitors.

Overall, our work gives exciting insights into the use of PARP and ATR inhibitors as a novel treatment strategy for HR-NB.

## Figures and Tables

**Figure 1 cancers-12-01095-f001:**
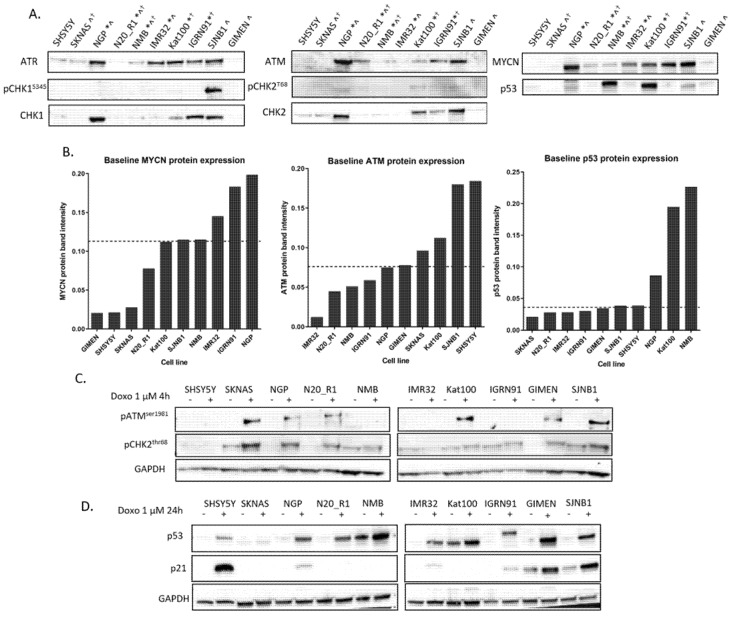
Expression and function of key DNA damage response (DDR) proteins in a panel of neuroblastoma (NB) cell lines. (**A**) Baseline protein expression of Rad3 related (ATR), ataxia telangiectasia mutated (ATM), CHK1, CHK2, phospho-CHK1^S345^, phospho-CHK2^T68^ MYCN and p53 in NB cell lines used in this study. Ponceau S stain was used as a loading control (Appendix A). *MYCN amplified, ^11q deleted, ^✝^TP53 mutant. (**B**) MYCN, ATM and p53 mean protein band intensity measured by densitometry (ImageJ) and normalised to total protein (Ponceau S) from 2 replicates. Cell lines are ordered by protein expression from low to high. The dashed line indicates median expression of each protein. (**C**) ATM function was determined by phospho-ATM^S1981^ and phospho-CHK2^T68^ expression after treatment with 1 μM doxorubicin (doxo) for 4 h. (**D**) p53 function was determined by p53 and p21 expression after treatment with doxo for 24 h. Images of the uncropped Western blots for A and C, D can be found in Appendix A

**Figure 2 cancers-12-01095-f002:**
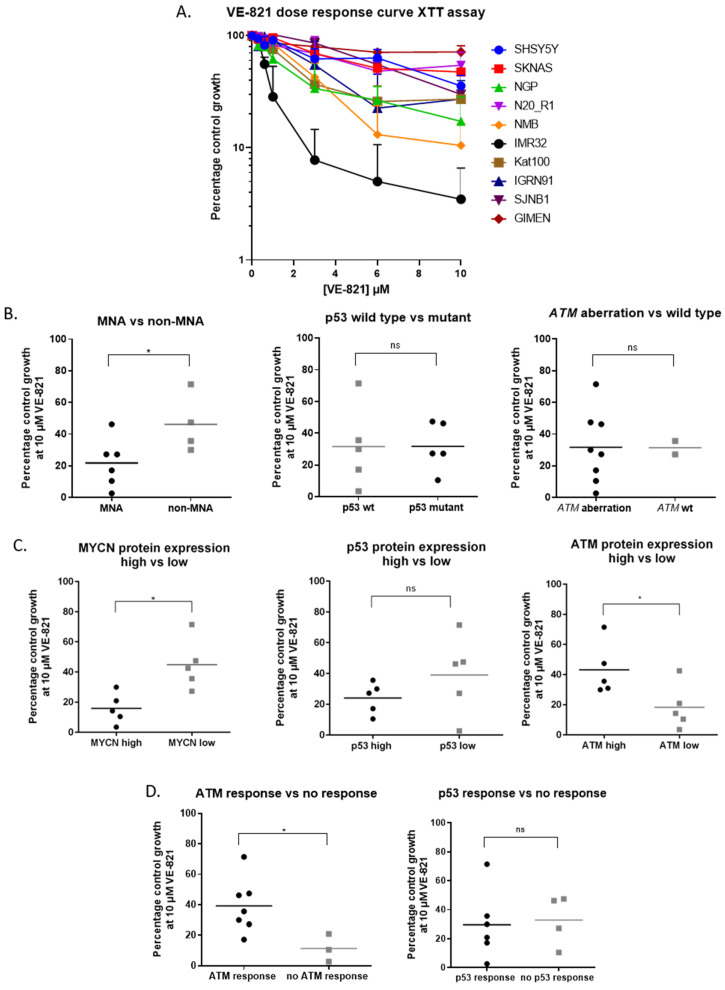
Determinants of ATR inhibitor sensitivity. (**A**) VE-821 (ATR inhibitor) dose response curves for NB cell lines, data are mean + SEM from 3 independent experiments. Cell lines were split into 2 groups based on (**B**) molecular features: *MYCN* amplification (MNA), p53 mutation status and ATM aberration (11q deleted or *ATM* mutated), (**C**) protein expression above (high) or below (low) median expression (Figure 1B) and (**D**) ATM and p53 responses after treatment with doxorubicin (1 µM). ATM and p53 response was determined by expression of pATM^ser1981^ and p21, respectively (Figure 1C,D). Average percentage control growth at 10 µM VE-821 was plotted for cell lines belonging to each group (*n* = 3). * *p* < 0.05 Mann Whitney U test, ns: not significant, wt: wildtype).

**Figure 3 cancers-12-01095-f003:**
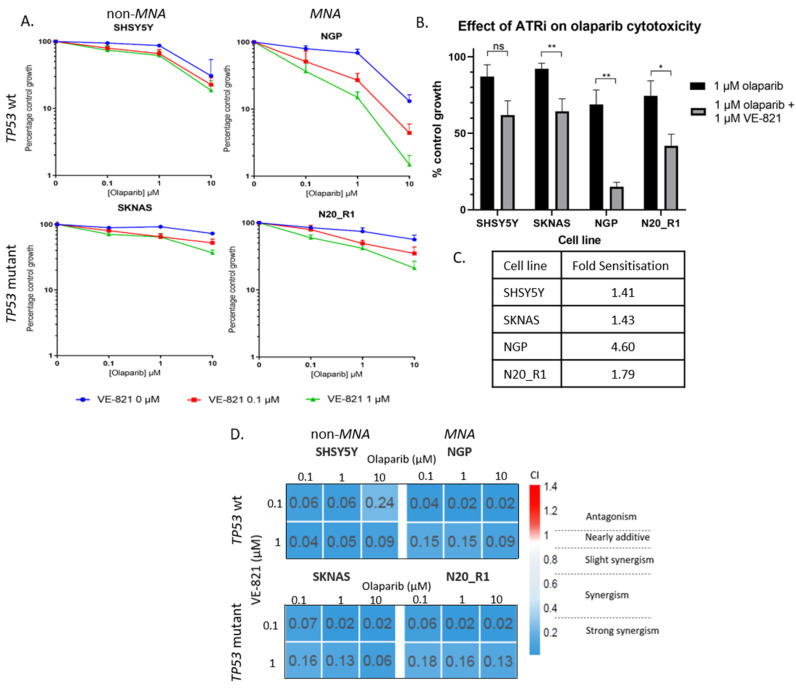
(**A**) XTT cell proliferation of the SHSY5Y, SKNAS, NGP and N20_R1 neuroblastoma cell lines treated with 0.1, 1 and 10 µM olaparib alone and with the addition of 0 (blue), 0.1 (red) and 1 (green) µM VE-821. Percentage control growth was normalised to effect of VE-821 alone. Data shown are the mean + SEM from 4 individual experiments. (**B**) Effect of 1 µM VE-821 on cytotoxicity of 1 µM olaparib normalised to the effect of VE-821 alone. Data shown are the mean + SEM from 4 individual experiments. T-test: ns; not significant, * *p* < 0.05, ** *p* < 0.01. (**C**) Fold sensitization of 1 µM olaparib by 1 µM VE-821 for each cell line. (**D**) Combination index (CI) values were calculated using CalcuSyn and plotted in heat map.

**Figure 4 cancers-12-01095-f004:**
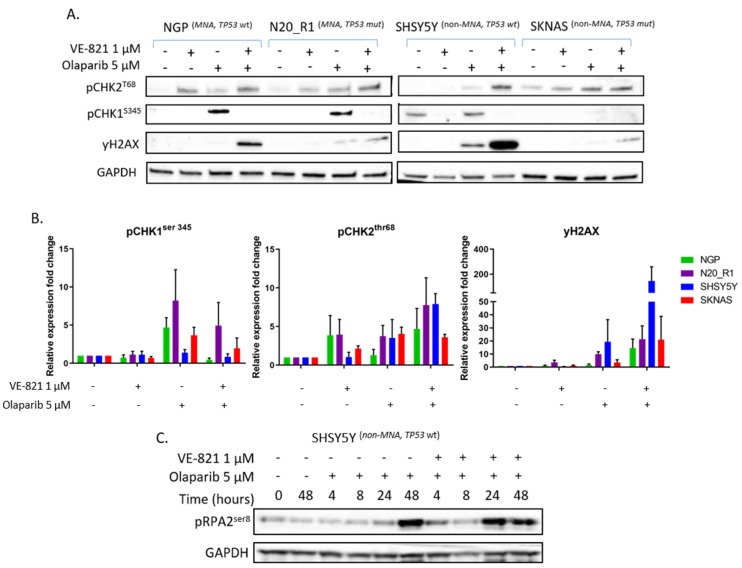
Effect of VE-821 and olaparib combination on replication stress. (**A**) pCHK1^S345^, pCHK2^T68^, γH2AX protein expression of NGP, N20_R1, SHSY5Y and SKNAS cells after incubation with 5 μM olaparib and/or 1 µM VE-821 for 24 h. (**B**) pCHK1^S345^, pCHK2^T68^, γH2AX mean fold change in protein band intensity measured by densitometry (ImageJ) and normalised to glyceraldehyde 3-phosphate dehydrogenase (GAPDH) loading control, data are mean + SEM from 3 independent experiments. (**C**) pRPA2^S8^ expression in SHSY5Y cell line in response to 5 μM olaparib with or without 1 µM VE-821 over 48 h. Images of the uncropped Western blots can be found in Appendix A.

**Figure 5 cancers-12-01095-f005:**
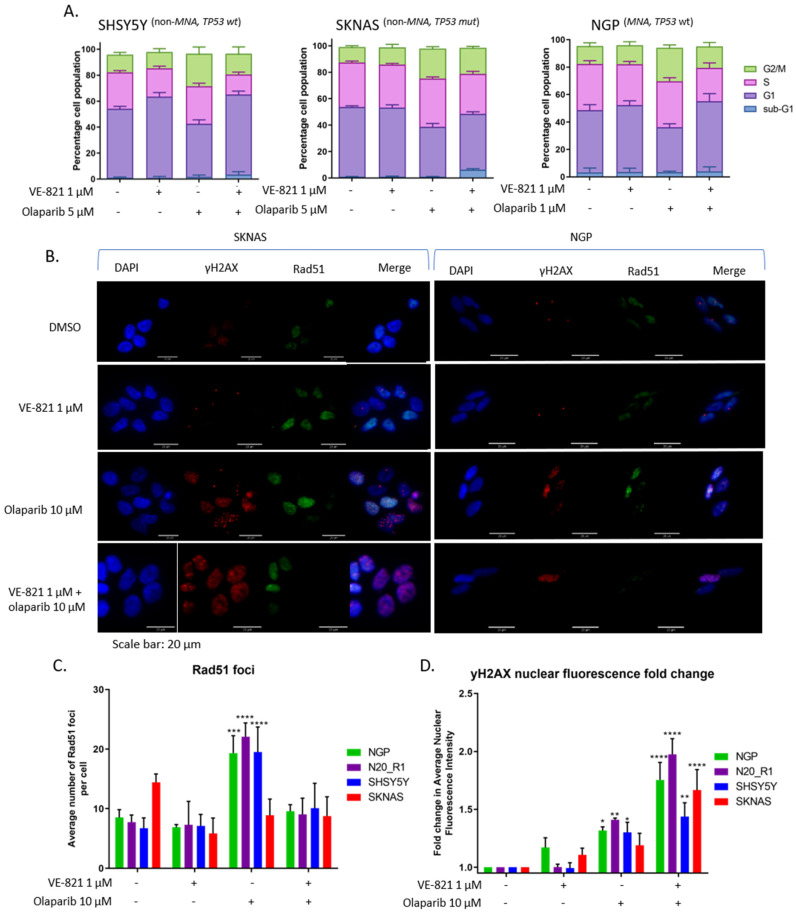
Effect of VE-821 and olaparib combination on cell cycle and homologous recombination repair (HRR). (**A**) Cell cycle analysis of SHSY5Y, SKNAS and NGP cell lines treated with 5 µM (SHSY5Y and SKNAS) or 1 µM (NGP) olaparib in combination with 1 μM VE-821 for 24 h. Data are mean + SEM from 3 independent experiments. (**B**) Representative γH2AX and Rad51 foci images from SKNAS and NGP cell lines treated with 1 µM VE-821, 10 µM olaparib or both for 24 h. (**C**) Average number of Rad51 foci per cell for NGP, N20_R1, SHSY5Y and SKNAS cell lines treated with 1 µM VE-821, 10 µM olaparib or both, *n* = 4 + sd. (**D**) Average γH2AX nuclear fluorescence intensity for NGP, N20_R1, SHSY5Y and SKNAS cell lines treated with 1 µM VE-821, 10 µM olaparib or both, fold change from control (DMSO), data are mean + SEM from 4 independent experiments. * *p* ≤ 0.05, ** *p* ≤ 0.01, *** *p* ≤ 0.001, **** *p* ≤ 0.0001, 2-way ANOVA difference from control (DMSO).

**Table 1 cancers-12-01095-t001:** Cell line genetic abnormalities.

Cell Line	*MYCN* Status	11q Status (Genes Deleted)	p53 Status	Reference
SHSY5Y	Non-amp	No deletion	WT	[34]
SKNAS	Non-amp	Deletion (MRE11, *ATM, CHEK1, H2AFX)*	Mutant Deletion of intron9/exon 10	[35,36]
NGP	Amp	Deletion (*ATM, CHEK1, H2AFX)*	WT	[36]
N20_R1	Amp	Deletion ** (*ATM, CHEK1, H2AFX)*	Mutant P98H P152T	[37]
NMB *	Amp	Deletion (MRE11, *ATM, CHEK1, H2AFX)*	Mutant G245S	[36,38]
IMR32	Amp	Deletion (*ATM, CHEK1, H2AFX)* ATM mutant V2716A	WT	[39,40]
IMR32/Kat100 (Kat100)	Amp	Unknown	Mutant C135F	[41]
IGRN91	Amp	No deletion	Mutant Duplication of exons 7–9	[42,43]
SJNB1 *	Non-amp	Deletion (MRE11, *ATM, CHEK1, H2AFX)*	WT	[36]
GIMEN	Non-amp	Deletion (MRE11, *ATM, CHEK1, H2AFX)*	WT	[36]

Amp: amplified, Non-amp: non-amplified, WT: wild type; * cell line is near tetraploid [44], Chr11 LOH; ** derived from NGP, assume same.

**Table 2 cancers-12-01095-t002:** Summary of VE-821 and olaparib single agent sensitivity.

Cell Line	*MYCN* Status	11q Status (Genes Deleted)	p53 Status	VE-821 GI50 (μM)	VE-821 LC50 (μM)	Olaparib GI50 (μM)	Olaparib LC50 (μM)
SHSY5Y	Non-amp	No deletion	WT	7.11	1.54	5.38	1.40
SKNAS	Non-amp	Deletion *(MRE11, ATM, CHEK1, H2AFX)*	Mut	>20	0.81	>30	1.31
NGP	Amp	Deletion *(ATM, CHEK1, H2AFX)*	WT	1.62	0.93	1.35	1.20
N20_R1	Amp	Deletion *(ATM, CHEK1, H2AFX)*	Mut	8.29	0.93	1.64	0.68
NMB	Amp	Deletion *(MRE11, ATM, CHEK1, H2AFX)*	Mut	2.36	1.91	3.88	0.92
IMR32	Amp	Deletion *(ATM, CHEK1, H2AFX) ATM mutant V2716A*	WT	0.66	0.90	1.81	0.63
Kat100	Amp	Unknown	Mut	1.88	1.50	>30	1.55
IGRN91	Amp	No deletion	Mut	3.04	1.03	>30	0.74
SJNB1	Non-amp	Deletion *(MRE11, ATM, CHEK1, H2AFX)*	WT	6.30	1.99	4.70	0.75
GIMEN	Non-amp	Deletion *(MRE11, ATM, CHEK1, H2AFX)*	WT	>20	0.87	12.25	0.37

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
