# Peer review of "ATR Inhibition Potentiates PARP Inhibitor Cytotoxicity in High Risk Neuroblastoma Cell Lines by Multiple Mechanisms"

_cancers, 2020, doi:10.3390/cancers12051095_

Round 1

Reviewer 1 Report

The authors have sufficiently justified the correlations, added much needed clarifications, and expanded their discussion of the presented data.  The additional array of statistical analyses were an unexpected bonus to their study.  The manuscript is now a significant and important study as it pertains to future HR-NB treatments.

Reviewer 2 Report

No additional comments 

This manuscript is a resubmission of an earlier submission. The following is a list of the peer review reports and author responses from that submission.

Round 1

Reviewer 1 Report

(1) Based on what evidence you hypothesized that combining ATR & PARP inhibitors can be synergistic?

(2) In figure 1.A, you showed CHK1 phosphorylation but no baseline expression of CHK1 was visible. How will you explain that?

(3) The figure legend isn't clearly describing the data presented in figure 2.B-D. Please rewrite the figure legend to help your reader better understand your observation.

(4) Clonogenic assay is generally performed to test the growth inhibitory effects not survival. Please share the raw images of your clonogenic assay.   

(5) Please mention which version of ImageJ software was used for the densitometric analysis of your western blots?

(6) You used olaparib at 10 micromolar & VE-821 at 1 micromolar dose in many of your experiments. Please provide rationale of using such doses.

(7) 10 micromolar of olaparib seems a very high dose for a kinase inhibitor. Can you make sure that at such high dose, olaparib isn't giving non-specific effects?

(8) Figure 5.A isn't clear; what do different parts of the column mean? Please make the figure more self-explanatory.

Author Response

Thank you for your positive comments and helpful suggestions.

Please see the attached document our responses and modifications to the manuscript.  

Reviewer 2 Report

Southgate et al. present a manuscript where they present potential biomarkers and treatments that may lead to novel therapy for high risk neuroblastoma, a devastating childhood cancer. Through the use of a multitude of high risk neuroblastoma cell lines, they analyzed the response of each cell line to treatment based on p53, myc, 11q (ATM/ATR) variations. Their findings suggest that high risk neuroblastoma that features myc amplification and ATR deletion can potentially be treated efficaciously with ATR+PARP inhibition.

This is a well-written manuscript and a study with many investigations and analyses, along with pertinent and sufficient statistical tests for each experiment. The problem is the lucidity of correlations when testing multiple variables (p53, MYCN, ATM, ATR, Chk1, Chk2, etc.). Protein levels of MYCN and ATM presented by the authors are inconsistent with the reported chromosomal deletions or mutations for several cell lines (see below). However, there is clear synergism when utilizing the ATM inhibitor + the PARP inhibitor, especially in the NGP line. With more justification of the correlations, via additional clarification and discussion of the presented data, the manuscript is potentially a significant and important study as it pertains to future HR-NB treatments.  No further experiments are needed; just more clarity on the explanations and justifications.

1.  Introduction: any information that the authors could include regarding the incidence or estimated number of cases of HR-NB per year would help with the significance

2.  Fig. 1A: Western blotting protein levels not in agreement with reported deletions/mutations:

-NGP has high Chk1 and ATM levels

-N20_R1 has significant levels of ATM

-SJNB1 has high MYCN, high ATM levels

Any discussion or insight into these findings would be helpful in justifying the correlations offered.

3.  Fig. 3A: this indicates synergism in all lines, but the efficacy of the combination treatments appears to be greatest in the NGP line. However, the statistical test (CI values) in 3B indicate slight synergism for 1 µM VE-821+10 µM olaparib in NGP cells. Can the authors explain this discrepancy?

4.  Fig. 5A: need to define bar graph components

Author Response

Thank you for your positive comments and suggestions. 

Please see the attached document our responses and modifications to the manuscript.  

Reviewer 3 Report

ATR inhibition potentiates PARP inhibitor cytotoxicity in high risk neuroblastoma cell lines by multiple mechanisms

Summary: The authors attempted to identify the molecular biomarker for ATR and PARP inhibitors in neuroblastoma and demonstrated that MYCN amplification and/or low ATM expression was associated to sensitivity to ATR inhibitor. They also found that sensitivity of ATR inhibitor could be synergetic with PARP inhibitor treatment.

Presentation of figure 1 is less clearly arranged. The order of cell lines in figure 1A is different from that of figure 1B. It is not clear whether the data of figure 1B is corresponding to that of figure 1A. ATM expression level in NGP is apparently the highest and SJNB1 is the second most. However, the graph in figure 1B fails to be consistent. Same critic is applicable to p53 level. It is necessary to shown mRNA expression level of MYCN, ATM and CIP1 (a downstream of TP53). In figure 1A, no protein loading was shown.

Total ATM protein level should be shown in figure 1C to see whether activity of ATM is corresponding to the level of ATM protein. The mutation information of each cell was poorly shown.

It is important to define MNA and non-MNA cell lines with clear evidence. MYCN amplification data and mRNA level along with protein expression would be necessary. Is mRNA expression level of MYCN corresponding to the efficacy of VE821? Linear regression would be useful.

It is not clear what ATM aberration means. How ATM aberration was defined? It is important to examine whether ATM protein level is corresponding to that of mRNA.

How ATM response was determined for figure 2D?

Efficacy of VE-821 was mostly determined by XTT data. Other than XTT, two other methods should be used to determine the efficacy (e.g. annexin V/7AAD, clonogenic assay and so on).

No statistical analysis in Figure 3.

Most of quantification was made based on the band density from the phospho-antibody. Instead, this should be validated by other analysis (e.g. flow cytometry).

It is not clear whether the difference shown in figure 4b is significant due to lack of statistics. Determination of band density is NOT a liable quantification analysis to lead any conclusion without other supporting data.

No statistical analysis in figure 5C so that what conclusion can be made based on the figures.

Minor comment

Discussion is unnecessary long and less organized.

Although the authors claimed that protein level of MYCN and activity of ATM could predict the sensitivity of ATR inhibitor, the data presented in the entire figures are very preliminary and less organized. The premature conclusion was drawn with unconvincing results. Quantification for efficacy and activity of ATM is very limited. No proper quantification analysis was performed to support their notion.

Author Response

Thank you for your suggestions. 

Please see the attached document our responses and modifications to the manuscript.  
